# Evaluation of the Microbiological Status of Cattle Carcasses in Mongolia: Considering the Hygienic Practices of Slaughter Establishments

**DOI:** 10.3390/vetsci10090563

**Published:** 2023-09-07

**Authors:** Munkhgerel Bayarsaikhan, Nyam-Osor Purevdorj, Byoung Hoon Kim, Jae Hun Jung, Gil Jae Cho

**Affiliations:** 1Department of Veterinary Public Health, School of Veterinary Medicine, Mongolian University of Life Sciences, Zaisan, Khan-Uul, Ulaanbaatar 17024, Mongolia; ub.vet.munkhgerel@gmail.com (M.B.); nyam-osor@muls.edu.mn (N.-O.P.); 2Institute of Zoonosis Infectious Diseases, College of Veterinary Medicine, Kyungpook National University, Daegu 41566, Republic of Korea; vetkbhoon@naver.com; 3Wosem Co., Cheongju 28114, Republic of Korea; jjh@wosem.co.kr; 4College of Veterinary Medicine, Kyungpook National University, Daegu 41566, Republic of Korea

**Keywords:** beef carcass, Enterobacteriaceae count, enterohemorrhagic *Escherichia coli*, HACCP, *Staphylococcus* spp. count, total bacterial count

## Abstract

**Simple Summary:**

Nomadic livestock husbandry is one of the key industries in Mongolia, with more than 70 million livestock, it can fully meet the domestic demand for livestock animal meat, and it has great potential for export to the international market. However, research on the hygienic quality of meat and the prevalence of foodborne pathogens is limited, and the efficacy of hygienic interventions applied in some establishments has not progressed. This is the first report on the evaluation of the baseline microbiological status of meat produced in Mongolian routine slaughterhouses with specific consideration of the efficacy of hygienic interventions.

**Abstract:**

The meat industry has received great attention in Mongolia, having over 70 million livestock, and is important to the nation’s economy. Systematic microbiological testing of carcasses has not been mandatorily regulated in all abattoir premises, and the efficacy of the introduction of the Good Hygiene Practice and Hazard Analysis Critical Control Points (HACCP) to some plants has not yet been tested microbiologically in Mongolia. Therefore, samples were collected from two establishments: plant A with an HACCP certificate from a third party and plant B without an HACCP certificate. The rates and levels of the total bacterial count (TBC) as overall hygiene indicators, the Enterobacteriaceae count (EBC) as fecal contamination indicators, and the *Staphylococcus* spp. count (SC) as personal hygiene indicators were determined on different parts of beef carcasses. The contamination rates in most parts were lower in plant A than in plant B (e.g., TBC in the rump and flank: 10^3^–10^5^ and 10^5^–10^7^, in plant A vs. 10^4^–10^6^ and 10^5^–10^8^ in plant B, respectively). Plant A also had a lower EBC and SC (*p* < 0.001). Furthermore, 2 out of 100 beef carcasses (2%) were positive for enterohemorrhagic *Escherichia coli* as a foodborne pathogen indicator in plant A.

## 1. Introduction

Livestock husbandry is one of the most important economic sectors in Mongolia. In 2022, Mongolia’s National Statistical Office recorded 71.1 million heads of livestock, including camels (0.7%), horses (6.8%), goats (38.8%), cattle (7.7%), and sheep (46%), which produced 13.2, 64.6, 97.7, 110.6, and 143.0 thousand tons of meat, respectively. In 2019, the average monthly consumption of meat and meat products per capita in Mongolia was 8.5 kg, whereas that in countries included in the Organization for Economic Cooperation and Development was 5.8 kg [1]. Up to 95% of the meat produced in Mongolia is consumed domestically, while only 5–9% is exported. Moreover, 90–95% of the meat produced in Mongolia is slaughtered and processed by small- and medium-sized businesses outside the formal abattoir system, with hygiene being largely ignored [2]. Given the poorly developed meat supply chain in Mongolia, complying with international standards is challenging. Artisanal slaughtering techniques are used in most of their domestically produced and consumed meat [3]. In 2013, the Code of Hygienic Practice for Meat (CAC RCP 58-2005, FAO/WHO Codex Alimentarius Commission) was adopted as a general hygienic rule for meat production in Mongolia. As of 2021, 119 slaughtering and meat processing plants were operating nationwide, mostly located in the capital and the central region. However, approximately 10% of the plants had implemented international standards such as the International Organization for Standardization (ISO) 22000, ISO 9000, and the Hazard Analysis Critical Control Point (HACCP) [1]. Indeed, meat is a highly perishable food that, unless correctly stored, processed, packaged, and distributed, spoils quickly and becomes hazardous because of microbial growth [4]. This condition is often the means through which foodborne illnesses may spread [5]. HACCP programs, good hygiene practices (GHP), and various interventions play a large role in enhancing the safety of food products. The effectiveness of these programs and interventions should be evaluated by conducting baseline studies to determine the levels of indicator organisms and the prevalence of pathogens [6]. Some of our preliminary study results (unpublished data) revealed that different slaughter facilities deal with different microbial loads on meat and have different understandings of implementing hygienic interventions such as HACCP and GHP. Additionally, no systematic research results are available to provide a basic background on the microbiological status of cattle carcasses and the effectiveness of hygienic interventions. Therefore, this study aimed to evaluate and compare the overall hygienic indicators (aerobic plate count and Enterobacteriaceae count), personal hygiene indicators (*Staphylococcus* spp.), and the presence of pathogen indicators (Enterohemorrhagic *Escherichia coli*, EHEC) between slaughter facilities with and without hygienic interventions, particularly HACCP.

## 2. Materials and Methods

### 2.1. Slaughter Establishments

Sample collection was conducted in two establishments, plant A and plant B. Plant A was an establishment with an HACCP certificate from a third body, and plant B was an establishment without an HACCP certificate. Plants A and B had similar conventional chilling systems. The abattoirs slaughtered between 5000 and 15,000 cattle annually. Slaughtering occurs from May to November, with up to 200 cattle carcasses per day.

### 2.2. Carcass Sampling

In this study, carcasses from native Mongolian cows aged 4–8 years and kept in natural pastureland were sampled. Samples were randomly collected after overnight chilling of carcasses, obtaining 33 and 62 carcass samples from plant A and B, respectively. Sampling was conducted independently, with five samples per day from each plant. The carcass swabbing areas were chosen according to the recommendations of ISO 17604, 2015. A 900 cm^2^ area was sampled for each carcass. The sampling sites were the rump, flank, and brisket, including the sternum. At each site, three independent 10 × 10 cm areas were sampled, giving a total of 300 cm^2^. The sites were swabbed using a premoistened sterile medical gauze cloth (Mesosoft, Mölnlycke Health Care, Gothenburg, Sweden) with 10 horizontal and 10 vertical movements. After sampling, the swabs from each site were pooled in one stomacher bag containing 300 mL of sterile peptone salt diluent and then kept refrigerated during transport to the laboratory. All samples were analyzed immediately on the day of sampling.

### 2.3. Bacteriological Analyses

**Hygiene indicators**: We determined the rates and levels of the total bacterial count (TBC) as the overall hygiene indicators, Enterobacteriaceae count (EBC) as fecal contamination indicators, and *Staphylococcus* spp. count (SC) as personal hygiene indicators. The stomacher bags containing swab samples were homogenized for 30 s using a stomacher (BagMixer 400 cc, Interscience Inc., Saint Nom, France). One ml of homogenates was taken and a 10^−8^ was conducted using peptone water. By using the pour plating method with standard agar (Nissui Pharmaceutical, Tokyo, Japan), colonies were counted after incubation at 37 °C for 24 h, and the number of bacteria was expressed as colony-forming units per cm^2^. Similarly, the EBC and SC were determined in appropriate dilutions of the serially diluted suspension by using deoxycholate agar (Nissui Pharmaceutical, Tokyo, Japan) and mannitol salt agar (Nissui Pharmaceutical, Tokyo, Japan) based on the colony characteristics, respectively. For the SC, we incubated the plates at 37 °C for 36 h and counted the golden yellow colonies as *Staphylococci*.

**Presence of pathogen indicators:** The presence of potential foodborne pathogens in cattle carcasses was determined by examining the prevalence of enterohemorrhagic *E. coli* (EHEC). We collected 100 random swab samples from three different parts of the cattle carcasses slaughtered in plant A, and these were independent from the abovementioned samplings. Sampling areas were covered 10 cm^2^ in each, with one cotton plug used for each area. The cotton plugs used for sampling on each carcass were pooled as one. Each pooled swab was incubated in 10 mL *E. coli* broth (mEC broth with novobiocin, Merk Millipore, Darmstadt, Germany) for 24 h at 37 °C. After this enrichment, 10 µL of the culture was inoculated into EHEC selective media (CT-SMAC, Nissui Pharmaceutical, Tokyo, Japan) for 18–20 h at 37 °C. The biochemical properties of the EHEC-like isolates were tested in Simmons Citrate, Triple Sugar Iron, and Lysine Indole Motility agars (Biolab, Budapest, Hungary).

### 2.4. DNA Extraction and PCR Analysis

The boiling method was used to extract bacterial genomic DNA from each EHEC-like isolate. Shiga toxin genes (Stx1 and Stx2) were amplified by PCR. The reaction mixture (20 μL) comprised 10 μL of Master Mix (AmpliTaq Cold 360, Thermo Fisher Scientific, Waltham, MA, USA), 8 μL of primer mix (primer sets for Stx1 and Stx2, WHO Collaborating Centre for Reference and Research on Escherichia and Klebsiella, 2010), and 2 μL of DNA sample. For the positive control, a DNA from previously validated isolate was used, and for the negative control, sterile MilliQ water was used. The reaction mixture was heated to 95 °C for 10 min, followed by 30 cycles, each consisting of 20s denaturation at 94 °C, 20 s annealing at 55 °C, 40 s extension at 72 °C, and a final extension of 3 min at 72 °C. The 10 μL PCR products were visualized on 2% agarose gels stained with ethidium bromide using electrophoresis, which was run at 100 V for 20 min.

### 2.5. Immunochromatographic Test (ICT)

Shiga toxin production by the PCR-positive isolates was determined using an ICT rapid test (NH immunochromato VT1/2, Cosmo Bio Ltd., Tokyo, Japan) according to the manufacturer’s instructions.

### 2.6. Statistical Analysis

Data analysis was performed using the SPSS statistical software program (SPSS for Windows version 16, SPSS Inc., Chicago, IL, USA). Counts of different bacterial loads per cm^2^ area at different sites of the cattle carcasses were transformed into log_10_ values prior to analysis and were expressed as the mean and standard deviation (SD). An independent *t*-test was used to detect the difference between the bacterial loads of the carcasses at two plants. The two-way analysis of variance (ANOVA) was used to compare the type of plant in terms of the carcass contamination rates in accordance with the sampled sites. The international limits used for comparison were sourced from Kim et al. [7].

## 3. Results and Discussion

Table 1 shows the TBC of the cattle carcasses from the two plants. Clearly, the bacterial contamination rate was lower in plant A. For instance, the TBCs in the rump and flank were 10^3^–10^5^ and 10^5^–10^7^ in plant A and 10^4^–10^6^ and 10^5^–10^8^ in plant B, respectively. These differences were also determined statistically (independent *t*-test: *p* < 0.001 for the rump and *p* < 0.05 for the flank) when analyzed as the log mean. This result suggests that HACCP intervention, at least, has a desirable effect on the contamination rate reduction (two-way ANOVA, *p* < 0.001). However, when compared with some international limits, such as the FAO (≤10^5^, Guideline, Food and Agriculture Organization, UN), Korean (≤10^5^, Guideline, Ministry of Food and Drug Safety 2014-135), and Australian (≤10^6^, Guideline, Meat Standard Committee, AS 4696:2007) limits, 36.4% and 9.1% of the brisket and flank sites from beef carcasses at plant A and 69.9% and 43.3% at plant B, respectively, exceeded the higher limit. Noticeably, the brisket site tended to have a wider count variation and a heavier bacterial load, particularly in plant B. However, the independent *t*-test, did not determine differences between the two plants, indicating a particular point to consider at the processing line in Mongolian practice and a critical control point to recognize when applying the HACCP system effectively.

As shown in Table 2, EBC as fecal contamination indicators were compared between the two selected plants. Again, plant A clearly had a good effect on the number of indicators, having a higher undetected rate than plant B (two-way ANOVA, *p* < 0.001). This statistical difference can be interpreted as the result of a higher variation (range) in the brisket site in plant B (*p* < 0.001), indicating the processing line point, where improvement is critically needed. Moreover, higher counts were detected when compared with the international limits such as the Korean (≤10^2^ CFU/cm^2^, Guideline, Ministry of Food and Drug Safety 2014-135), U.S. (≤10^2^ CFU/cm^2^, Standard, Food Safety and Inspection Service, Electronic Code of Federal Regulations), EU (≤10^2^ CFU/cm^2^, Standard, Commission Regulation, No 2073/2005), Australian (≤10^3^ CFU/cm^2^, Guideline, Meat Standard Committee, AS 4696:2007), and FAO (≤10^3^ CFU/cm^2^, Guideline, Food and Agriculture Organization, UN) limits.

Table 3 lists the range of the SC from different sites of cattle carcasses at the two plants. In general, both the range and mean values of the SC clearly differed between the plants. Plant A had smaller ranges and fewer counts than plant B. Given this dissimilarity, the statistical tests identified significant differences according to the sampled sites and plants. *Staphylococci* in meat are often related to poor hygienic practices during processing [8], and unsanitary conditions in slaughterhouses or contamination from the handler’s skin, mouth, or nose can be a major cause of high bacterial prevalence [9]. Thus, *Staphylococci* could be one of the potential target organisms for improving the hygienic quality of meat in consideration of personnel-derived indicators and critical control point identification in the successful implementation of HACCP in Mongolian plants.

Enterohemorrhagic Escherichia coli (EHEC) is a Shiga toxin producing *E. coli* associated with the most severe forms of foodborne illnesses [10]. Serogroup O157 predominates over other EHEC serogroups, but the isolation frequency of non-O157 EHEC has increased slightly over the past few years [11]. EHEC can cause hemorrhagic colitis and hemolytic uremic syndrome in humans [12]. Cattle are natural carriers of Shiga toxin-producing *E. coli* [13], and they generally do not show pathological symptoms in their natural hosts [14,15]. However, equipment, particularly knives, saws, and tables, used in the beef carcass production line can further become a vector to spread Shiga toxin-producing *E. coli* onto other carcasses and cuts of meat [16]. Thus, considering hygienic intervention efficacy, the EHEC surveillance on beef can be one of the most important targets for meat safety. In this study, 100 beef carcasses were sampled for the screening of EHEC and pathogen prevalence. Based on the growth in selective enrichment broths and colony characteristics on selective agar plates, 46 isolates were obtained initially. Thereafter, the isolates were tested against gas, acid, and H_2_S production in Triple Sugar Iron media; the presence of lysine decarboxylase, indole production, and motility in Lysine Indole Motility media; and citric acid utility in Simmons Citrate media biochemically. In the end, 12 isolates were obtained as EHEC-like isolates, and the DNA was extracted to detect Shiga toxin genes by PCR. Figure 1 shows the PCR result for the Shiga toxin genes.

According to the PCR results, Shiga toxin genes were detected in 2 out of 12 EHEC-like isolates microbiologically. Both of these cases belonged to plant A. As shown in Figure 1, the PCR-positive isolates possessed the Stx1 gene (lanes 4 and 5) and Stx2 gene (lanes 6 and 7), independently. Furthermore, toxin production in these isolates was confirmed by ICT. Figure 2 depicts the ICT result of Stx1 production by a PCR-positive isolate as a representative. Based on these results, the prevalence of EHEC in beef can be estimated as 2% (2/100) and is comparable to those found in carcasses of culled dairy cow 1% [17] and feedlot cattle 6% [18]. HACCP systems at meat processing plants should be based on microbiological data, with estimates of the numbers of indicator organisms on meat products at various stages of processing [19]. The results of the present study indicated that hygienic intervention such as HACCP in plant A has an effect on the reduction in the TBC, EC, and SC when compared to plant B. The two plans were not substantially different in factors influencing meat hygiene other than HACCP. Thanks to the flexibility of the HACCP, the system proves to be effective regardless of the business context in which it is applied. To validate and verify the proper implementation of HACCP principles, conducting a detailed study identifying or diagnosing points to be improved is possible based on indicators such as those used in this study. With the preliminary data from this study, in consideration of the limitations of accessible and appraisable data, the importance of the implementation of HACCP, which has not been mandatorily regulated for meat processing plants, and the failures in validation such as the verification and corrective actions for proper implementation of the systems is revealed as in plant A. There were higher counts than international limits, which strongly suggests the need for further improvements in Mongolia. More importantly, there is a serious national institutional lack in the accreditation of food safety interventions [20], suggesting that it is necessary to implement HACCP under the leadership of the government on a national level rather than plants’ autonomous approaches. It may be efficient to benchmark hygiene and safety policies with similar sentiments from developed countries, such as Korea.

## 4. Conclusions

In Mongolia, initiatives for HACCP implementation in slaughter plants have emerged. The results of this study suggest that the contamination rates (TBC, EBC, and SC) were reduced in plants with HACCP implementation. However, further improvements are essential to align successful HAACP applications. Moreover, to improve the status of the microbiological safety of meat at the abattoir level, the government should address a nationwide policy on HACCP.

## Figures and Tables

**Figure 1 vetsci-10-00563-f001:**
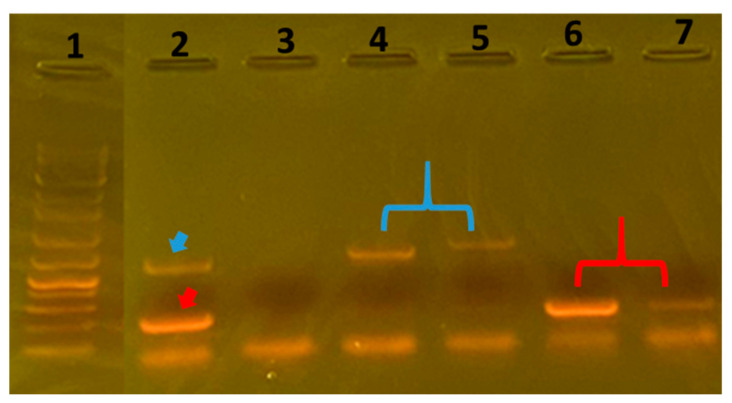
Stx genes detection. Lane 1, DNA marker; Lane 2, positive control for Stx1 (red arrow) and Stx2 (blue arrow); Lane 3, negative control; Lanes 4 and 5 (technical repetitions), Stx1 gene of the EHEC; Lanes 6 and 7 (technical repetitions), Stx2 gene of the EHEC.

**Figure 2 vetsci-10-00563-f002:**
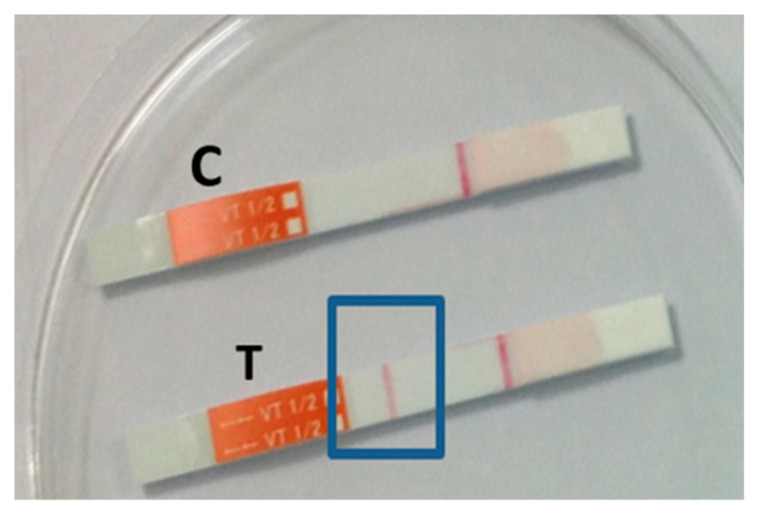
Production of Stx1 by EHEC. C, control stripe; T, test stripe; the blue box encloses the positivity of Stx1.

**Table 1 vetsci-10-00563-t001:** Frequency of the rates and levels of TBC (A) and the statistical comparison (B).

A							B					
Range, CFU/cm^2^	Plant A, n = 33			Plant B, n = 62			Variables	Plant A, n = 33		Plant B, n = 62		Independent *t*-Test
	Rump	Brisket	Flank	Rump	Brisket	Flank		Mean	SD	Mean	SD	*p* Value
≤10^3^	0.0	9.1	0.0	0.0	3.3	0.0	Rump	3.92	0.50	4.79	0.64	<0.001
10^3^–10^4^	54.5	0.0	0.0	10.0	0.0	0.0	Brisket	4.66	1.14	5.26	0.90	ns
10^4^–10^5^	45.5	9.1	0.0	43.3	3.3	3.3	Flank	4.48	0.33	4.95	0.74	<0.05
10^5^–10^6^	0.0	45.5	90.9	46.7	23.3	53.3	ANOVA	ns		ns		
10^6^–10^7^	0.0	36.4	9.1	0.0	53.3	30.0		Two-way ANOVA				
10^7^–10^8^	0.0	0.0	0.0	0.0	13.3	13.3	Plant types	<0.001				
Above detection	0.0	0.0	0.0	0.0	3.3	0.0	Sites	<0.05				
	100.0	100.0	100.0	100.0	100.0	100.0	Plant type x Sites	ns				

Legend: ns—statistically not significant at *p* ≥ 0.05.

**Table 2 vetsci-10-00563-t002:** Frequency of the rates and levels of EBC (A) and the statistical comparison (B).

A							B					
Range, CFU/cm^2^	Plant A, n = 33			Plant B, n = 62			Variables	Plant A, n = 33		Plant B, n = 62		Independent *t*-Test
	Rump	Brisket	Flank	Rump	Brisket	Flank		Mean	SD	Mean	SD	*p* Value
Undetected	9.1	27.3	18.2	3.3	13.3	0.0	Rump	1.35	1.06	1.91	0.76	ns
≤10^1^	45.5	45.5	36.4	13.3	3.3	6.7	Brisket	1.11	0.67	2.31	0.92	<0.001
10^1^–10^2^	27.3	18.2	27.3	46.7	33.3	60.0	Flank	1.42	0.95	1.80	0.59	ns
10^2^–10^3^	9.1	9.1	9.1	30.0	33.3	30.0	ANOVA	ns		<0.05		
10^3^–10^4^	9.1	0.0	9.1	6.7	6.7	3.3		Two-way ANOVA				
10^4^–10^5^	0.0	0.0	0.0	0.0	6.7	0.0	Plant types	<0.001				
Above detection	0.0	0.0	0.0	0.0	3.3	0.0	Sites	ns				
	100.0	100.0	100.0	100.0	100.0	100.0	Plant type x Sites	ns				

Legend: ns—statistically not significant at *p* ≥ 0.05.

**Table 3 vetsci-10-00563-t003:** Frequency of the rates and levels of the SC (A) and the statistical comparison (B).

A							B					
Range, CFU/cm^2^	Plant A, n = 33			Plant B, n = 62			Variables	Plant A, n = 33		Plant B, n = 62		Independent *t*-Test
	Rump	Brisket	Flank	Rump	Brisket	Flank		Mean	SD	Mean	SD	*p* Value
≤10^1^	18.2	18.2	0.0	6.7	0.0	0.0	Rump	1.97	0.89	3.73	0.81	<0.001
10^1^–10^2^	27.3	45.5	45.5	13.3	0.0	23.3	Brisket	1.73	0.64	3.62	0.75	<0.001
10^2^–10^3^	45.5	36.4	54.5	40.0	16.7	43.3	Flank	2.04	0.45	3.54	0.78	<0.001
10^3^–10^4^	9.1	0.0	0.0	26.7	30.0	23.3	ANOVA	ns		ns		
10^4^–10^5^	0.0	0.0	0.0	3.3	26.7	3.3		Two-way ANOVA				
10^5^–10^6^	0.0	0.0	0.0	10.0	3.3	6.7	Plant types	<0.001				
Above detection	0.0	0.0	0.0	0.0	23.3	0.0	Sites	ns				
	100.0	100.0	100.0	100.0	100.0	100.0	Plant type x Sites	ns				

Legend: ns—statistically not significant at *p* ≥ 0.05.

## Data Availability

All data supporting the findings of this study are available from the corresponding author upon request.

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
