# Peer review of "Evaluation of the Microbiological Status of Cattle Carcasses in Mongolia: Considering the Hygienic Practices of Slaughter Establishments"

_vetsci, 2023, doi:10.3390/vetsci10090563_

Round 1
Reviewer 1 Report
Line 34 – What is NSO 2021? Isn’t that a reference? Why isn’t that the ref no. 1?
Line 52 - Food safety is to consumers, regulatory agencies, governments, and the food industry.? I don’t get the meaning.
Line 65 – Remove this (E.)
Line 91-92 – Reword this sentence - …and tenfold serial dilution up to……..
Lines 93-95 – Simply mention that you followed pour plating with XX agar.
Line 107 - choosing different carcasses. – What do you mean over here?
Line 109 - in 10 E. coli broths_ Is that 10 ml?
Line 113 – Can you provide a better citation - (all from Biolab, Hungary
Line 115 – Did you follow any previous protocols? Where is the citation? Read a couple of good articles to check how to word the PCR conditions. What are the conditions for electrophoresis? What are the controls (positive and negative)?
Line 175 - Plant A had smaller ranges and fewer counts plant B?
Figure 1 – Are lanes 4 and 5 your samples? I got the answer after reading the text. But, change your figure legend. Positive control does not contain both stx 1 and 2?
Line 211 – Have you compared 2% prevalence with other studies? Is that similar or different?
Line 206-242 – Contain lot of repeating information.
There are differences between the 2 plants. But it is hard to tell the difference is due to HACCP and GHP without sampling more plants, although it is a costly process. In this case, EHEC was isolated from the plant with HACCP.
References are numbered twice in my version.
It is hard to understand some sentences. Please work on that. Otherwise your idea is not clearly understood.
Author Response
We revised it as your requested. I'd appreciate it if you could review the attachment.

Reviewer 2 Report
Overall this is a very interesting and useful paper. However, the sample size is a bit low. There was no indication of throughput at the abattoir, neither was there an indication of the age, breed and quality of cattle slaughtered. Minor problems with English - its not really acceptable to start a sentence with an acronym.
Should not start sentences with an acronym.
Author Response

(The authors gave the same response as above.)
